# Interferometric Instrument for Thickness Measurement on Blown Films

**Michele Norgia** * and **Alessandro Pesatori**

Politecnico di Milano, Dipartimento di Elettronica e Informazione e Bioingegneria (DEIB),
Piazza Leonardo da Vinci 32, 20133 Milano, Italy; alessandro.pesatori@polimi.it
* Correspondence: michele.norgia@polimi.it

**Abstract:** Real-time measurement of plastic film thickness during production is extremely important to guarantee planarity of the final film. Standard techniques are based on capacitive measurements, in close contact with the film. These techniques require continuous calibration and temperature compensation, while their contact can damage the film. Different optical contactless techniques are described in literature, but none has found application to real production, due to the strong vibration of the films. We propose a new structure of low-coherence fiber interferometer able to measure blown film thickness during productions. The novel fiber-optic setup is a cross between an autocorrelator and a white light interferometer, taking the advantages of both approaches.

**Keywords:** low-coherence interferometry; thickness measurement; optical coherence tomography; plastics measurement

## 1. Introduction

Plastic films are used in countless applications, from normal bags, food containers, packaging or other varied uses. The manufacturing technique is based on blown film extrusion [1], realized through large cylindrical bubbles (1–2 m of diameter), see, for example, the photo in Figure 1. At the top of the system, the bubble is cut to be then wound on a roller. Through the dosers, it is possible to manage the composition of the plastic film, while its thickness depends on the height of what is called the "freeze-line": the point where the plastic solidifies and the bubble stops expanding.

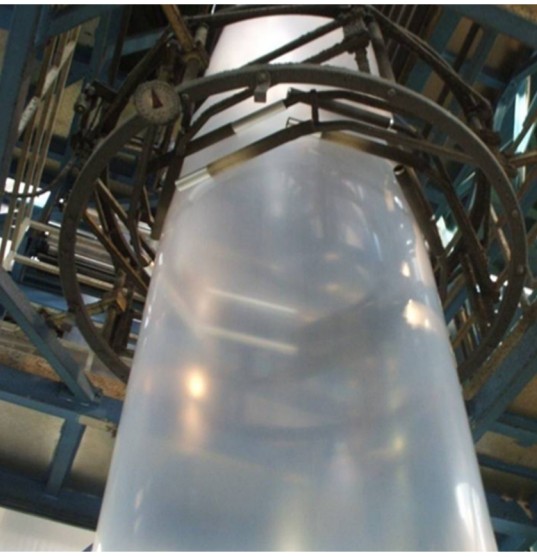

**Figure 1.** Photo of a bubble for plastic film extrusion. Bubble diameter is 1.5 m.

To realize a flat film, without swelling, the thickness of the bubble must be extremely uniform. In addition, there are numerous applications where a minimum thickness should be guaranteed, that is why the temperature at the beginning of the bubble is controlled accurately through a number of valves opening a jet of cold air. The thickness control cannot work properly in open loop, so a real-time measurement performed during the bubble growth is essential. Online gauging systems are used to provide automatic feedback control on the valves. Considering the bubble extrusion, it is impossible to access the inner film face during the extrusion, therefore, the thickness measurement must be obtained just from one side. This requirement excludes some standard techniques, such as absorption methods of light [2] or radiations [3], or simple distance measurements from both sides [4]. The classic technique from one side is based on capacitive gauging systems. The bubble is scanned using a circular slide and the sensor must keep a deep contact with the film to allow capacitive sensing [5]. This approach, even if extensively adopted, is limited by some problems. First of all, it gives a relative measurement because of troubles in holding calibration. To obtain the absolute thickness value, if the density is known, the gravimetric control is the normal choice. Considering that the film is quite hot, the change of temperature after the touch of the film induces a transient which is very difficult to compensate. Another problem arises with multilayers of film: if one layer has a dielectric constant quite higher than the others, the capacitive sensor calibration is lost if the percentages of the different layers vary. Finally, contact measurement often leaves a slight mark on the film during its movement, and for some film applications, it could be unsightly.

Some non-contact techniques are proposed for film gauging from one side, such as optical confocal methods or laser triangulation [2]. These techniques allow to have source and detector on the same side of the plastic film, however, the low intensity with which the beam focused on the second interface reaches the photodetector often does not allow to distinguish the signal from the noise of the measurement system. Other optical methods are based on low-coherence interferometry [6], a well-known technique used for Optical Coherence Tomography (OCT) [7–11]. In this paper, we present a modified fiber-optic low-coherence interferometer, able to work directly on a plastic bubble during extrusion, thanks to high-speed scanning and high depth of measurement field.

## 2. Materials and Methods

A low-coherence interferometer is based on the interference of an optical source, characterized by a limited coherence: the interference happens only when two beating optical beams have a path difference lower that the coherence length $L_c$ [6]. For light sources with a Gaussian spectrum, the coherence length is equal to:

$$L_c = \frac{2\,\ln(2)}{\pi} \cdot \left( \frac{\lambda_0^2}{\Delta\lambda} \right)$$

(1)

where $\Delta\lambda$ is the spectral linewidth of the optical source and $\lambda_0$ is its central wavelength. In classical low-coherence interferometer, the reference arm is constituted by a movable mirror to perform the spatial scanning: as the optical path difference varies, there will be interference between the recombined beams and, consequently, the classical function of periodic interference with sinusoidal trend. Since the source coherence is low, the fringes have a periodicity $\lambda/2$ as in a normal interferometer, but they are modulated in amplitude by the coherence function. For a source with Gaussian spectrum, also the coherence function is Gaussian. Figure 2 shows the intensity measured at the receiver of the output branch of a Michelson interferometer [2] when a source with coherence length $L_c$ is used to measure the thickness $d$ of a plastic film, with refractive index $n$, and the reference mirror is moved at speed $v$. The delay measurement between the pulses with Gaussian envelopes is a measurement of the thickness $d$ of the plastic films, obtained by Equation (2):

$$d = \frac{v}{n}\bar{t},$$

(2)

where $\bar{t}$ is the time distance between the two pulses.

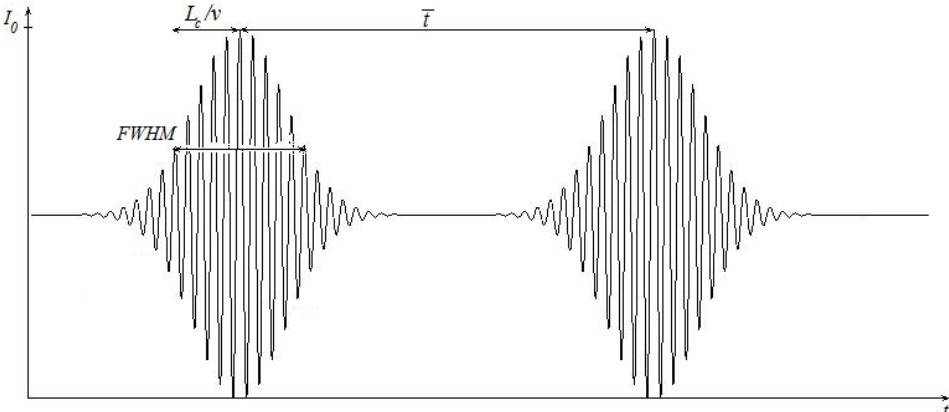

**Figure 2.** Photodetector interference figures in a Michelson interferometer with a low-coherence source when measuring a plastic film.

As evident from Figure 2, the coherence length limits the spatial resolution of the thickness measurement.

For measurement on real bubble for blown film, we decided to implement an all-fiber configuration for the low-coherence interferometer. Figure 3 shows the scheme of the first prototype realized for this particular application.

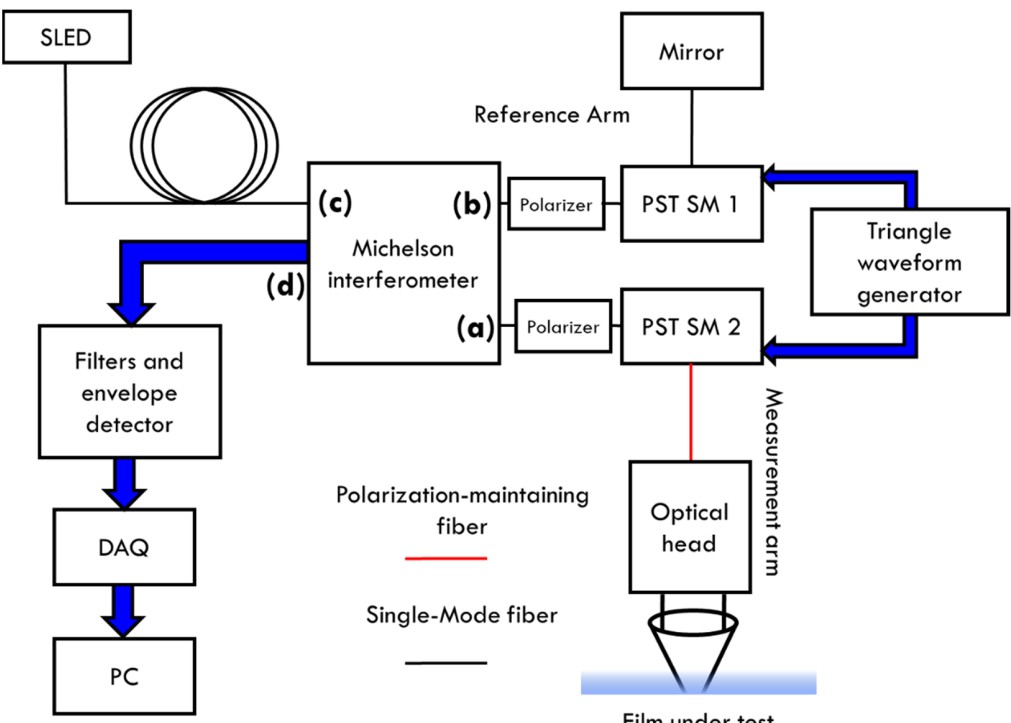

**Figure 3.** Scheme of the first prototype of low-coherence interferometer. SLED is superluminescent diode; DAQ is Data Acquisition card; PST SM is Piezo Stretcher with Single Mode fiber; PC is Personal Computer: (**a**) is measurement output; (**b**) is reference output; (**c**) is SLED input; (**d**) is electrical output.

The low-coherence light source (EXS1320-2111, EXALOS) is a superluminescent diode (SLED) with optical fiber output, at $\lambda_0 = 1315$ nm, $\Delta\lambda = 60$ nm corresponding to about

12 μm of coherence length. It is fed to an all-fiber Michelson interferometer (model INT-MSI-1300, THORLABS). Its scheme is described in Figure 4.

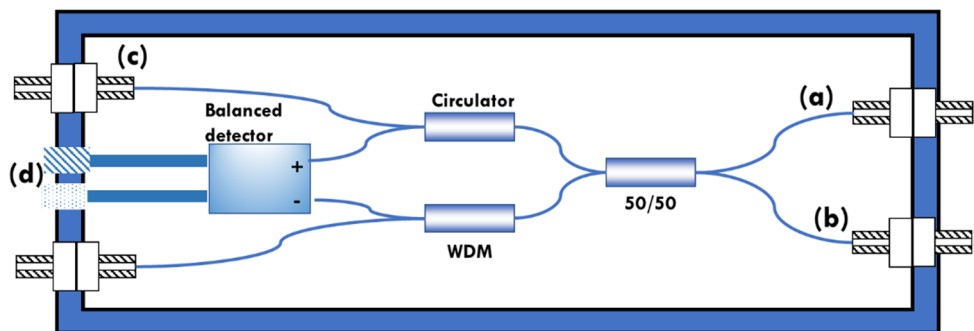

**Figure 4.** Scheme of the commercial all-fiber Michelson interferometer. (**a**) is measurement output; (**b**) is reference output; (**c**) is SLED input; (**d**) is electrical output.

The circulator couples the optical power incident at circulator ports, through the directional coupler (50/50), into the measurement (a) and reference branches (b). The reflections of the two branches recombine in the coupler and are directed to the two photodiodes. The interferometer makes the difference between the photodiodes signals, implementing a balanced detection. Compared to the classic solution, this system with circulator exploits doubles the power, in addition to considerably reducing the common mode components and avoiding back-reflection to the source, very dangerous for a superluminescent diode. The interferometer is also equipped with a wavelength division multiplexer (WDM) for an input in the visible region (a red laser typically), to better see the measurement spot.

The reference branch is equipped with a fiber stretcher (PZ2 Fiber Stretchers, OPTIPHASE) to modulate the length of the optical path, in order to perform a spatial scan along the axis of the measuring branch without having to physically move the mirror of the reference branch. Other scanning methods are available, but they require the movement of some optical and mechanical element, thus reducing the reliability and average life of the device, especially in the case of continuous operation of the system. Other than this, we experimentally found that the scanning frequencies should be higher than 150 Hz, to minimize the errors induced by film vibrations on real bubbles. This scanning rate is difficult to realize with a mechanical mirror. Fiber stretcher is driven by a triangular wave generator and a subsequent high voltage amplifier. After traveling through the two branches, the optical signal reflected returns to the inside of the interferometer where the interference is acquired by a photodetector. The driving voltage of the fiber stretcher must be carefully studied, to obtain a spatial scanning which allows the observation of the interference fringes due to the two plastic film interfaces: air-plastic and plastic-air. It must also be taken into account that any non-linearity in the scan translates into a non-linearity of the measurement. The equivalent speed of the mirror of the reference branch, which depends on the driving parameters of the fiber stretcher, determines the frequency of the fringes and the bandwidth of the electrical signal output from the interferometer. In fact, the following relationships hold true:

$$f_{fringes} = \frac{2 \cdot v_{mirror}}{\lambda_0}, \tag{3}$$

$$\Delta f = 2 \cdot v_{mirror} \frac{\Delta \lambda}{\lambda_0^2}, \tag{4}$$

where $f_{fringes}$ is the frequency of the fringes of the signal leaving the interferometric block, $\Delta f$ is the width of its frequency spectrum, $\cdot v_{mirror}$ is the equivalent speed of the mirror on the reference branch, $\lambda_0$ is the central wavelength of the light source and $\Delta \lambda$ is the width of the emission spectrum.

For the prototype made, the PZ2 model of Optiphase was chosen as fiber stretcher, which includes both the piezoelectric cylinder and the fiber suitable for wavelengths between 1200 nm and 1600 nm. It can be controlled with voltages between ±400 V with a fiber extension factor of 3.8 μm for each applied volt. Considering then that the refractive index of the optical fiber is equal to 1.45, there is an effective variation of the optical path of about 5.5 μm for each applied volt. This value is not exact, because when the fiber is stretched, the effective refractive index decreases correspondingly, that is why the system will need a calibration, described in Section 3.1. From the electric point of view, it exhibits a strong resonance at 18 kHz; for lower frequency, it can be modeled as a purely capacitive 100 nF load. It is necessary to operate at frequencies much lower than 18 kHz to avoid oscillations due to resonance. It is also important to avoid abrupt transitions in the driving voltage, so as not to stimulate spontaneous oscillations of the system.

Figure 5 shows the periodic modulating wave, digitally designed to have a linear scan on both directions, but smoothed in order to reduce drastically its high-frequency harmonics. The voltage is generated by the analog output of a microcontroller, with main frequency of 180 Hz.

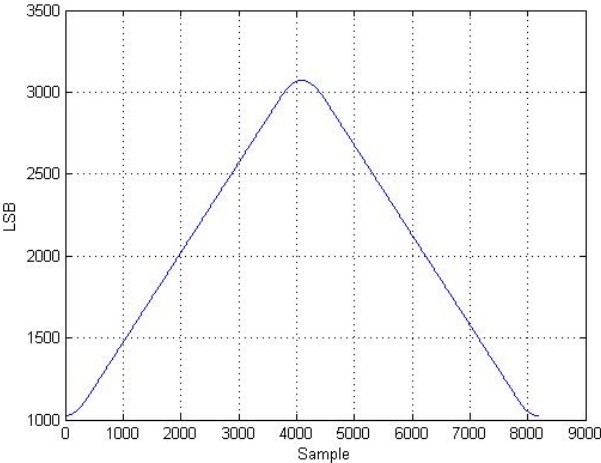

**Figure 5.** Modulating wave, digitally generated with smoothed edges to not exciting fiber stretcher piezo resonances (18 kHz). LSB is the number of Less Significant Bit, read by the microcontroller to generate the output wave.

The modulating wave is finally amplified at high voltage by a custom-made amplifier. Figure 6 shows the driving wave applied to the fiber stretcher, attenuated by a 1/31 divider.

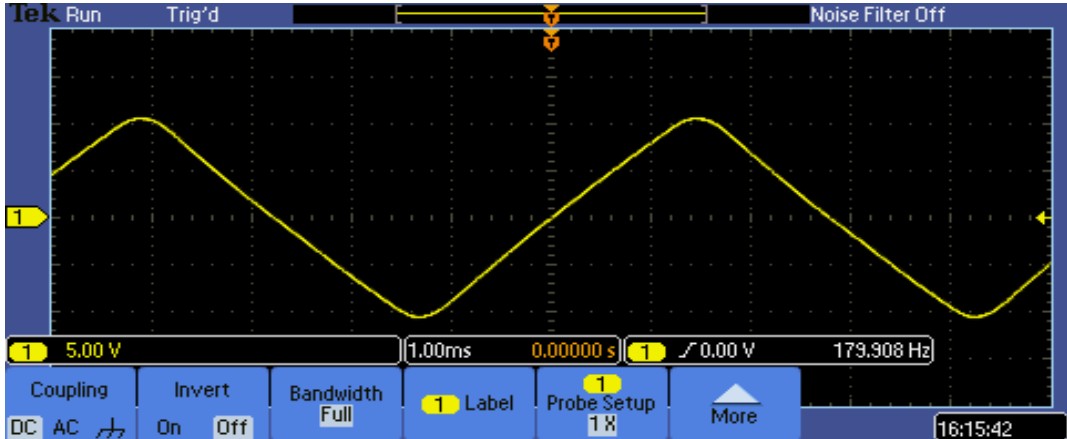

**Figure 6.** Detail of the waveform output from the amplifier and applied to the fiber stretcher, scaled by a factor of 1/31. Amplitude scale: 5 V/division. Horizontal scale: 1 ms/division.

Figure 7 shows its spectrum; the cancellation of harmonics above 2 kHz is evident.

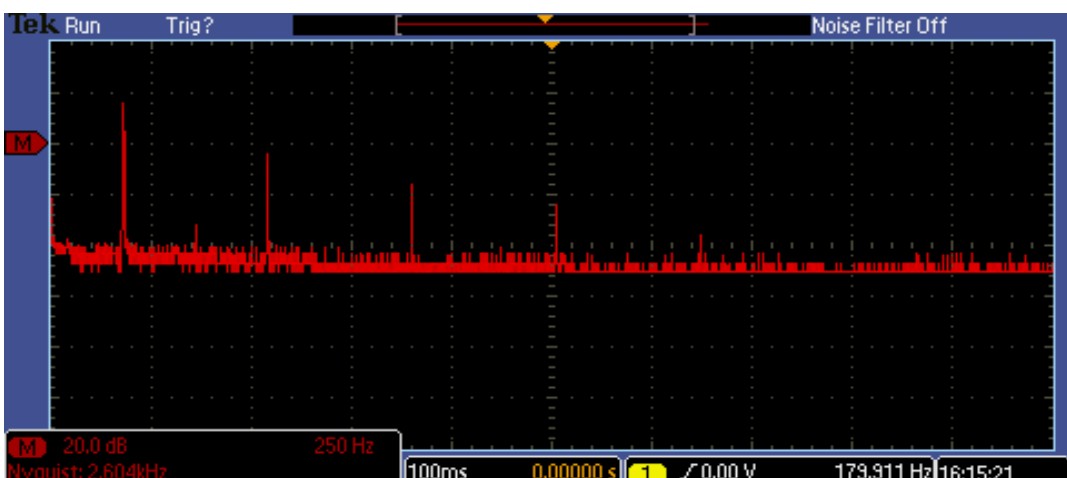

**Figure 7.** Spectrum of the signal applied to the fiber stretcher. All disturbances above 2 kHz are canceled to avoid fiber stretcher resonances. Amplitude scale: 20 dB/division. Horizontal scale: 250 Hz/division.

The output signal from the Michelson interferometer consists of interference fringes with a Gaussian envelope. The frequency of these fringes, given by Equation (3), is equal to about 2.4 MHz. To filter at this frequency, an active second order band-pass filter was chosen, inspired by band-pass configuration of the Sallen–Key cell. Next, electronics is an envelope detector and a final low-pass filter. The output signal is acquired by a DAQ-card (model NI 6251) and processed by a PC via LabVIEW.

The last section of the instrument is the optical head, connected through a polarization maintaining fiber, to minimize polarization variation while moving the head. Light is collimated and focused on the plastic film, as shown in Figure 8.

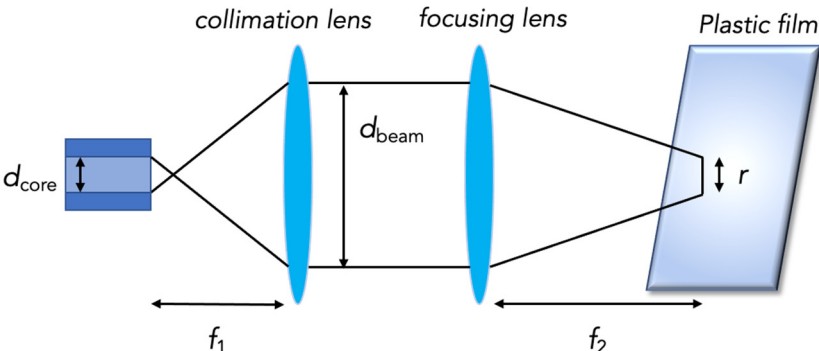

**Figure 8.** Optical head: collimation optics and beam focusing.

The collimator is the F240APC-C model by Thorlabs, working at $\lambda$ = 1310 nm, with a focal length $f_1$ = 8 mm and a numerical aperture of 0.50. In this way, as indicated by the collimator specifications, a collimated beam of about 1.4 mm in diameter is obtained. It is focused by a biconvex lens with focal length $f_2$ = 25 mm, thus obtaining a transversal resolution $r$ equal to:

$$r = \frac{4\lambda_0}{\pi} \frac{f}{d_{beam}} = \frac{4 \times 1310 \text{ nm}}{\pi} \cdot \frac{25 \text{ mm}}{1.4 \text{ mm}} \cong 30 \text{ µm}, \tag{5}$$

where $d_{beam}$ is the diameter of the light beam after its collimation. As can be seen from Figure 8, the transverse resolution, $r$, represents the width of the beam focused on the measurand. The choice of these lenses made it possible to obtain a depth of focus $D_{focus}$

of about 1 mm. The depth of focus is the distance from the focus point, where average intensity has decreased by a factor 2, and it is given by:

$$D_{focus} = \frac{\pi r^2}{2\lambda}.$$ (6)

With the depth of focus of 1 mm, the instrument can provide acceptable signals within $\pm 2$ mm of dynamics. This means that it is sufficient to keep the surface of the bubble within these at 4 mm around the focus distance $f_2 = 25$ mm to obtain a correct measurement. From a practical point of view, this means that the measuring system requires a relatively slow bubble follower. The focal length chosen for the focusing lens represents a good compromise between retroreflected power collected and depth of focus obtained. Another factor to be considered is the sensitivity to the incident angle of the light, that in theory should be perpendicular to the film surface. With a focal length of 25 mm and a collimated beam diameter of 1.4 mm, there are $\pm 1.6$ degrees of tolerance for the alignment, to still achieve some reflected signal. The error in thickness measurement for this kind of misalignment is absolutely negligible (it is a cosine error lower than $10^{-3}$): the problem related to misalignment is only signal fading; if the sensor can see the signal, the measurement value is correct. Finally, the distance between the collimating lens and the focusing lens must be such that the overall length of the measuring branch equals that of the reference branch, to obtain the maximum output signal when the fiber stretcher is in the rest condition.

The last part of the first prototype consists in two polarization controllers, placed in both measurement and reference arm, to change the light polarization in the fiber, to avoid a different path for the two main polarizations in the fiber. If not well controlled, the two paths of different polarizations generate 2 measurement peaks for every surface, significantly worsening the resolution of the instrument. This control in laboratory condition is easily obtained manually, but it is not stable in the real environment, due to strong temperature variations (plastic bubble is at high temperature, and ambient temperature around the bubble can easily exceed 40 degrees).

The problem of light polarization for these kind of interferometers is well known. A standard solution is given by a different scheme, the auto-correlator [12], shown in Figure 9. This scheme measures the beating frequencies between the reflection of the two surfaces, instead of the ones between a mirror and a single surface. It has the great advantage to not require two measurement branches of the same length and it is insensitive to polarization effect, thanks to the Faraday mirrors. The main drawback is the requirement of strong light reflection from the target surface because internal mirror light reflection is much higher than the surface reflection. This requirement forces to work only with a strong focusing of the measurement beam on the target, and it makes it not possible to reach an instrument depth of focus of 4 mm.

In order to keep the advantages of the low-coherence interferometer (Figure 3) adding the absence of sensitivity to polarization, typical of auto-correlator, we propose a novel scheme, shown in Figure 10. The main difference from the traditional one, Figure 3, is the addition of a Faraday mirror on the reference branch, and a Faraday rotator on the measurement branch, close to the optical head. Another difference is the use of a single fiber stretcher, which proved sufficient for the application, considering 4 mm of depth of focus.

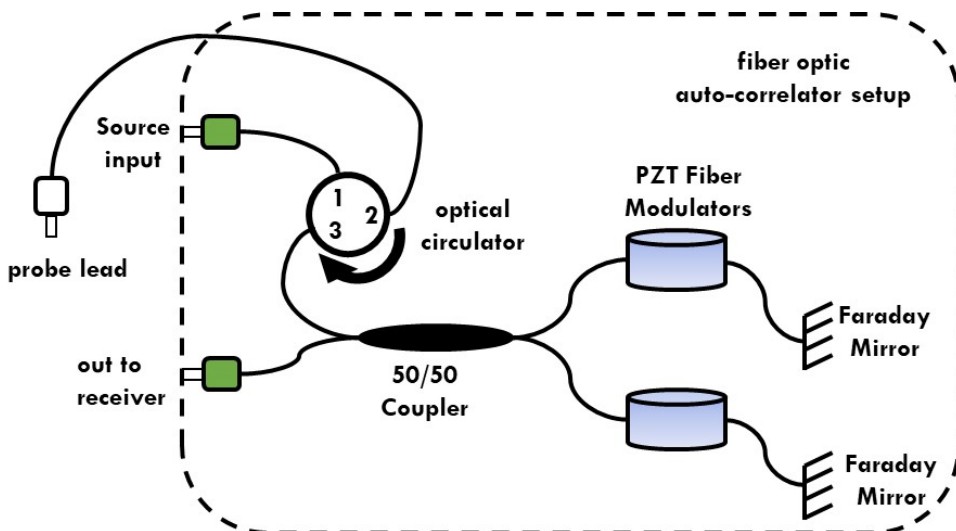

**Figure 9.** Scheme of an all-fiber autocorrelator for thickness measurement: the beating happens between the reflections from the two film surfaces.

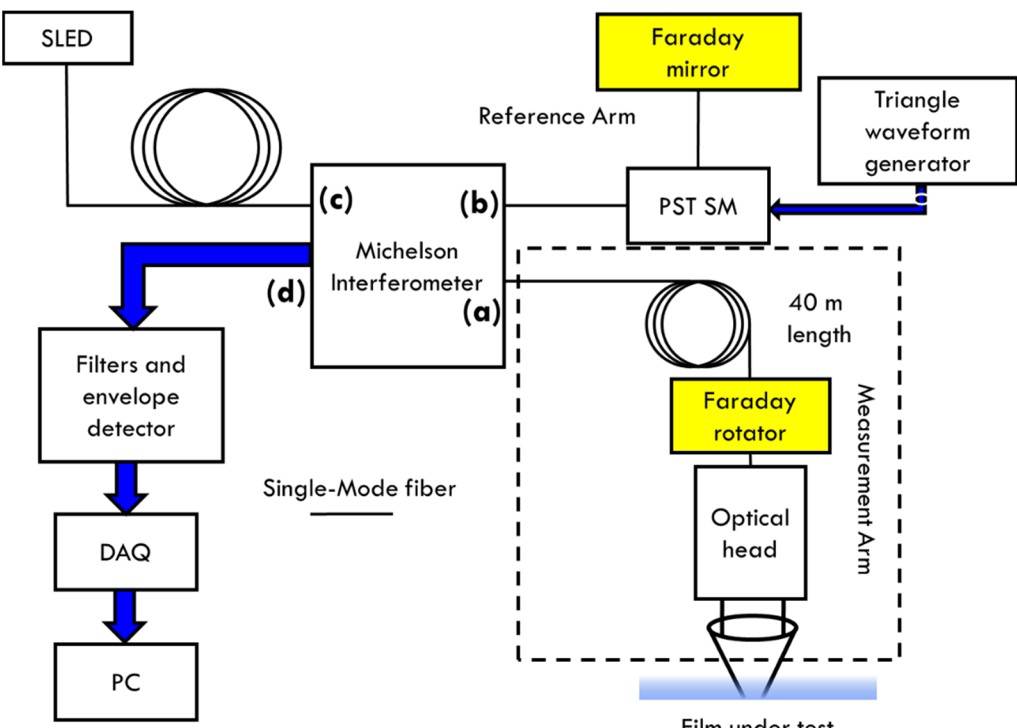

**Figure 10.** Final set-up of low-coherence interferometer, with Faraday mirror and rotator, to solve polarization problems. SLED is superluminescent diode; DAQ is Data Acquisition card; PST SM is Piezo Stretcher with Single Mode fiber; PC is Personal Computer. (**a**) is measurement output; (**b**) is reference output; (**c**) is SLED input; (**d**) is electrical output (same labels of Figures 3 and 4).

Faraday rotator is an optical device that rotates the polarization of the light flowing in it by exploiting the Faraday effect, the result of a ferromagnetic resonance [13]. This resonance causes the decomposition of the waves into two rays with reverse circular polarization which propagate at different speeds (circular birefringence). Due to the difference in propagation speeds, at the end of the medium, the two rays recombine with a sharp phase difference which results in a rotation of the angle of the polarization plane. The light exiting Faraday rotator will be rotated 45° with respect to the input polarization.

Subsequently, this light will reach the plastic film and will be reflected. Passing again through the Faraday rotator, it will be rotated by another 45° so that the return light is orthogonal to the outward one. In this way, the polarization changes along the fiber are canceled during the return journey, because for every input polarization state, the sum of the round trip remains a constant path. The same happens for the Faraday mirror, inserted in the reference branch, able to compensate for the polarization problems, also due to the birefringence induced by the fiber stretcher [14].

The final prototype shows a complete insensitivity to polarization problems, maintaining the excellent performance of the first prototype. Figure 11 shows an example of an acquired signal while measuring a transparent film, together with the triangular modulating wave.

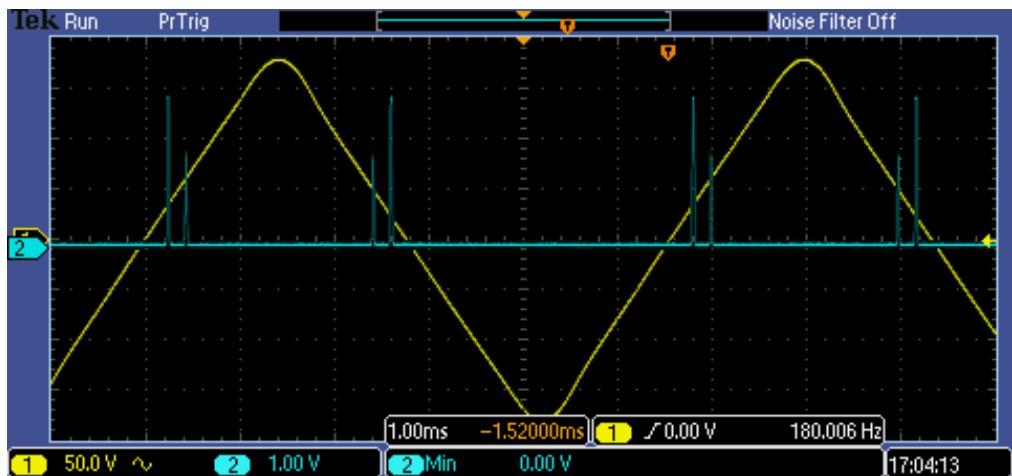

**Figure 11.** Acquired signal (light blue signal) while measuring a transparent film, together with the smoothed triangular modulating wave (yellow signal). Amplitude scale: 50 V/division for triangular wave; 1 V/division for acquired signal. Horizontal scale: 1 ms/division.

## 3. Results

The prototype instrument elaborates acquired signals for obtaining the more accurate thickness measurement. Figure 12 shows an example of raw signal acquired while measuring a transparent plastic film, with nominal thickness of 50 μm. In the figure, the two couple of peaks are evident, corresponding to the ascendant and descendant phases of the triangular modulation. Time distance between two peaks is proportional to the film thickness.

With regards to the amplitude variability of every peak, we have to note that the peak's amplitude depends on the angular alignment of the sensor, on the parallelism of the two surfaces, and also on the absolute distance (the relative position with respect to the focus). For these reasons, in every measurement, we can see different amplitudes, which are also relative between the peaks corresponding to the two surfaces. To pass from time measurements to thickness measurement, an instrument calibration is required, as described in the later paragraph.

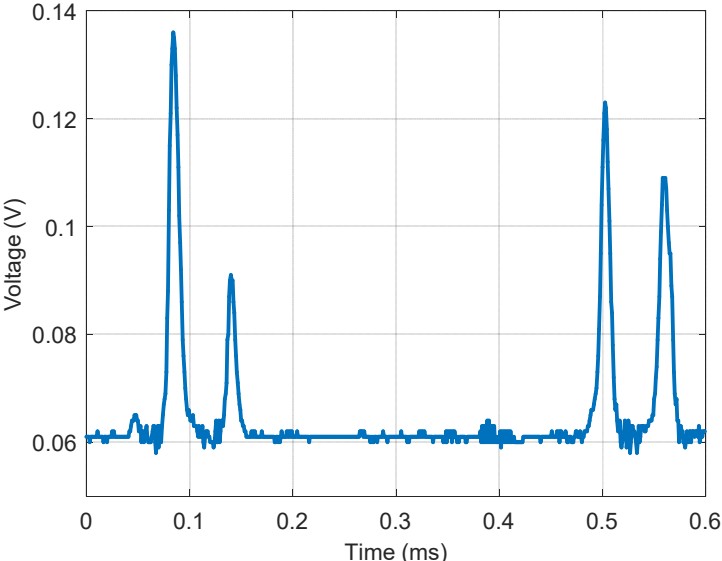

**Figure 12.** Raw signal acquired for a transparent film; nominal thickness 50 μm.

### 3.1. Calibration

The calibration of the instrument was carried out using a single interface target (aluminum wall), moved by a micrometric slide. In this way, the instrument measures only one peak. The calibration curve is reported in Figure 13: the final range is higher than 4 mm, linearity is guaranteed on all the range and the sensitivity in air is about 1.5 mm/ms. When measuring plastic films, the refractive index $n$ should be considered. For the types of plastic normally employed (for example, polyethylene), it is around 1.5.

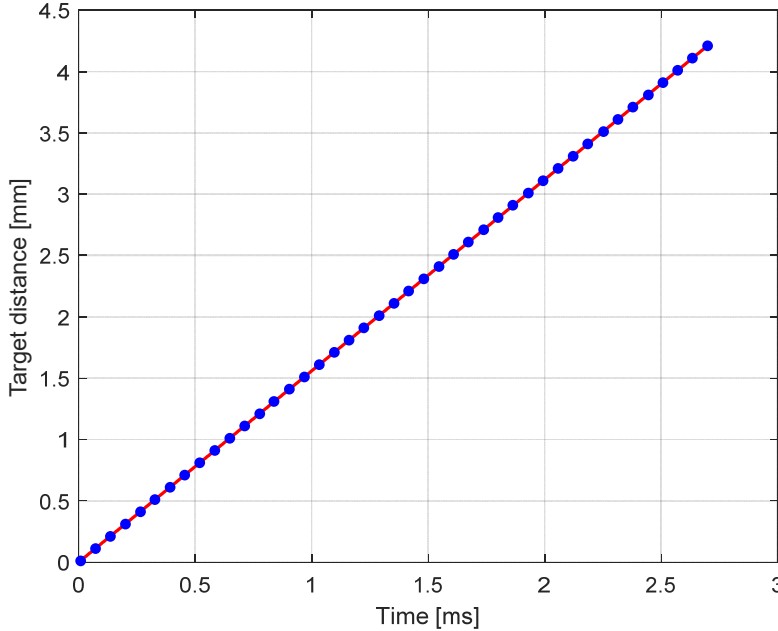

**Figure 13.** Calibration of low-coherence interferometer (in air). Dots represent measured data while the line is the linear fitting curve. The measurement range of about 4 mm is centered at 25 mm from the output lens (see Figure 8).

### 3.2. Average Algorithm

The algorithm for calculating film thickness requires an input signal as clean as possible, without disturbances caused by bubble vibrations or by the presence of dyes.

For this reason, 10 forward spatial scans are acquired before evaluating the thickness. Subsequently, these scans are aligned since the first peak (recognized with a parabolic regression technique), and then averaged to obtain a signal as independent as possible from the vibrations. The same procedure is carried out with the return spatial scans.

In Figure 14a, we see the trend of 10 forward scans aligned based on the first peak. Due to the vibrations, they will not all be perfectly alike; however, the distance between the peaks, which indicates the thickness of the plastic film, will be kept constant. By averaging the 10 scans, we can eliminate noise and disturbances, as we can see in Figure 14b.

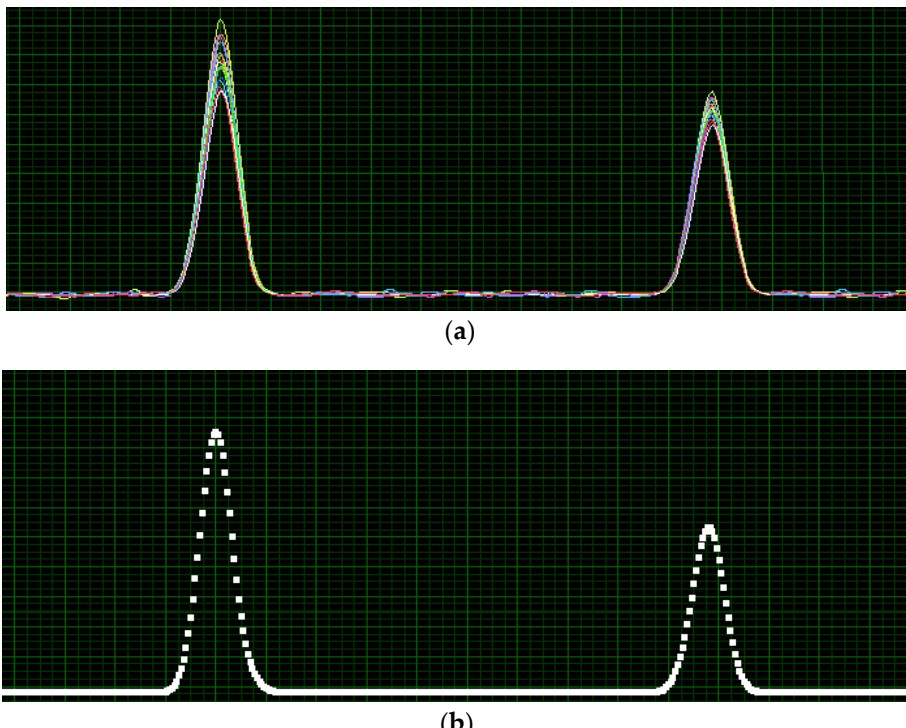

(**a**)

(**b**)

**Figure 14.** (**a**) Ten subsequent acquisitions, aligned on the first peak. Amplitude scale: 100 mV/division. Horizontal scale: 10 μm/division; (**b**) Average of the acquisitions from Figure 14a. Amplitude scale: 100 mV/division. Horizontal scale: 10 μm/division.

Several algorithms have been tested for the best localization of the peaks: optimal filters, correlation with Gaussian functions, center of gravity. It was noted that the best accuracy and repeatability performances were obtained with a parabolic regression algorithm, applied to 8 points around the maximum. This algorithm has the advantage of being extremely simple and can be easily implemented even on a microcontroller: the analytical solution of linear regression is applied to the derivative of the curve, and the zero of the signal derivative corresponds to the maximum of the peak.

### 3.3. Measurement on A Real Plant

The instrument was initially mounted on a fixed location in front of the bubble. The normal vibrations of the bubble do not exceed the 4 mm range of the instrument, so it was possible to carry out a continuous measurement of the film thickness. A very first version of the instrument, modulated at 20 Hz, did not exhibit a stable measurement, while the actual prototype, at 180 Hz, show a series of measurements that move in the measurement range, but once realigned on the first peak, they show a variation of a few micrometers, practically negligible after the 10 averages. Figure 15 shows the two measures during rise and fall of the triangular wave, for 80 μm transparent film. Horizontal scale is already calibrated in micrometers. The measured peaks now are not perfectly symmetric, because the vibration changes the peak position (the scanning speed is much higher than

the vibration speed, but there is still an influence). By averaging 10 measurements, this effect is strongly reduced, but there is still a contribution on the measurement accuracy: the measured standard deviation is about 0.1 µm in laboratory conditions, while on real bubble, it is between 1 µm and 2 µm, depending on the bubble vibration.

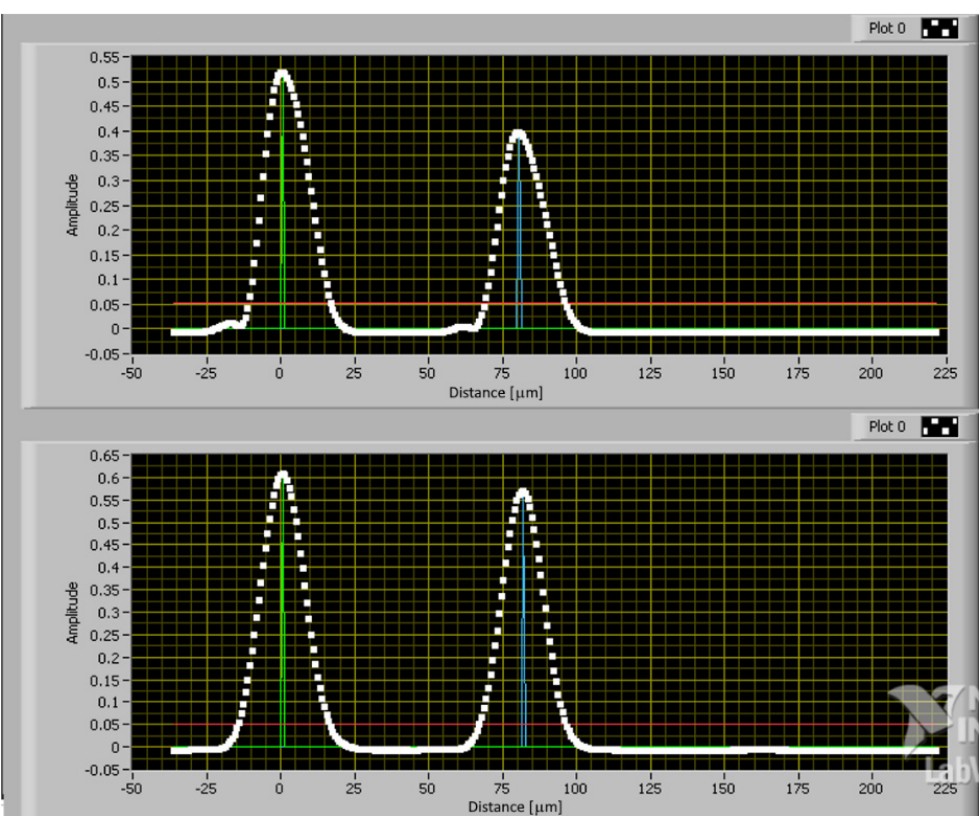

**Figure 15.** LabVIEW screenshot of the measurement acquired on a real bubble in a working plant, for a transparent film with thickness 80 µm, during rise (**top**) and fall (**bottom**) of the triangular wave. Amplitude scale: 50 mV/division. Horizontal scale: 25 µm/division. The two vertical lines indicate the position of the peaks, localized by the parabolic regression algorithm.

After a series of positive results on real bubbles, the instrument was mounted on the rotating structure of a commercial capacitive sensor. In this case, the measurement is easier because the contact sensor dampens the natural vibrations of the bubble. In any case, comparable measurement results were obtained by releasing the capacitive sensor (bubble free to vibrate). The comparison of thickness measurement from capacitive and optical sensors is reported in Figure 16: Figure 16a shows the measurement for a transparent film with nominal thickness of 80 µm, as a function of the angle of the rotating stage; Figure 16b shows the measurements acquired during a change of film thickness. The capacitive sensor was automatically disconnected during the change. The small thermal drift of its measurement while touching the film (at angle 180°) is also evident.

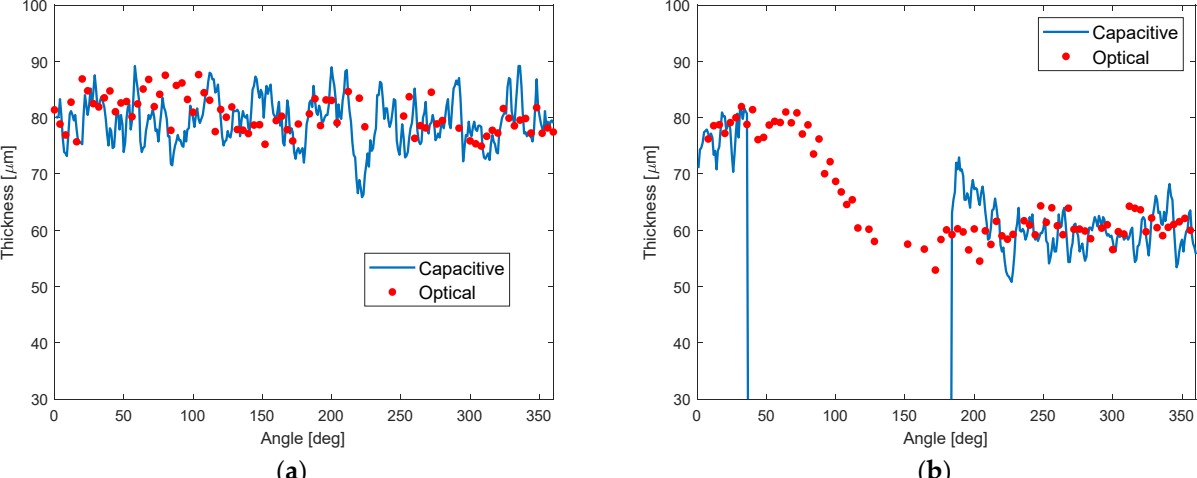

**Figure 16.** Measured thickness during rotation around the bubble. Comparison of capacitive and optical measurements: (**a**) Constant thickness; (**b**) Measurement during a change of film thickness, from 80 μm to 60 μm.

## 4. Discussion

The proposed instrument for contactless thickness measurement is based on a modified low-coherence interferometer. It is realized in an all-fiber configuration, in order to take advantage of the high modulation speed of piezo stretcher, mandatory for measuring on a real bubble, vibrating at a frequency of a few hertz. In order to overcome the polarization problems induced by the optical fiber, standard techniques for thickness measurement are based on autocorrelation: the beating interference is given between the reflections of the two surfaces of the film. This technique exhibits good results in laboratory conditions, but it is not adequate to application in a real plant, because it requires strong focusing on the film, in order to get enough back-reflection. In this way, the depth of focus is too limited to include bubble vibrations. Our proposal is based on a standard low-coherence interferometer, enriched by two faraday rotators, able to compensate for the fiber birefringence, even if time-variant. To our knowledge, this configuration is original. The designed depth of focus is about 4 mm, coherent with the scanning range. From a measurement campaign on a real plant, the range proved to be adequate to compensate for bubbles' vibrations, considering the measurement frequency of 180 Hz.

With reference to performance, the instrument has a resolution lower than 1 μm, limited by the coherence length of the source (about 12 μm) improved by the parabolic regression on the peaks. The standard deviation, at a measurement rate of 180 Hz, in laboratory condition is about 0.1 μm, while on real bubble, it is between 1 μm and 2 μm, depending on the vibration conditions (function of the film thickness). Minimum measurable thickness is about 20 μm. As expected from theory, we cannot see any measurement error due to incident angle variation, with an angle tolerance of about ±1.6 degrees to still obtain some reflected signal: the error is a cosine function of the angle and it is negligible for this misalignment (lower than $10^{-3}$). The main problem of misalignment is signal fading: on real bubble, the sensor should be carefully aligned to be perpendicular to the film surface, by looking at the signal amplitude. Experiments have shown rapid distance variations due to bubble vibrations, but small angular misalignment. On tens of measurement on real plants, we have noticed no signal fading due to misalignment.

In conclusion, the proposed instrument is a possible substitute for capacitive sensors, as these performances are adequate for the specific measuring application of plastic films.

**Author Contributions:** Conceptualization, M.N. and A.P.; methodology, M.N. and A.P.; software, A.P.; writing—original draft preparation, M.N.; writing—review and editing, A.P. All authors have read and agreed to the published version of the manuscript.

**Funding:** This research received no external funding.

**Institutional Review Board Statement:** Not applicable.

**Informed Consent Statement:** Not applicable.

**Conflicts of Interest:** The authors declare no conflict of interest.

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
