# Peer review of "Interferometric Instrument for Thickness Measurement on Blown Films"

_photonics, doi:10.3390/photonics8070245_

Round 1

Reviewer 1 Report

The authors proposed an instrument to measure the thickness of blown films by using white light interferometry. The film thickness is demodulated with the position difference of the two interferometric peaks that correspond to the reflected light from the film's front and rear surface. The peak position is determined by the position of its envelope maximum. This instrument is very useful in the plastic film manufacturing factory. Before published in MDPI Photonics, I recommend the authors respond to the comments and questions below.

  1. Figure 1 is a photo of a bubble for plastic film extrusion. What's the size of it? I recommend the authors provide the scale of the photo.
  2. The authors should explain the acronyms again in the figure caption of Figure 3, i.e. SLED, PST SM 1/2, DAQ, PC.
  3. Is the modulating wave generated by an arbitrary wave generator? What's the function expression of the wave? In addition, what is the meaning of the y-axis label of Figure 6?
  4. Figure 7 and Figure 8 show the screenshots of the graphic user interfaces of a Tek oscilloscope (OSC) and frequency spectrum analyzer (FSA), respectively. I think it is OK for a thesis paper. But we seldom put software screenshots in a journal article. I don't have the same OSC and FSA as yours. So, I cannot easily read out the amplitudes of the wave or the frequencies of the harmonics. Actually, Figures 12, 15, and 16 have a similar problem.
  5. Figure 9 is too blurry and omits the plastic film under test. Thus, the authors avoided answering the problem that how to assure the incident angle of the light is perpendicular to the film surface.
  6. The peak amplitudes in Figure 13 are not consistent with those in Figure 12. From the light blue signal in Figure 12, we can see that the amplitudes of the first and the fourth peaks are approximately equal and the amplitudes of the second and the third peaks are approximately equal. However, we cannot see this phenomenon in Figure 13.
  7. Figure 14 should have a legend to describe that the blue dot is measured data while the red line is the fitting result.
  8. Figure 16 should have a legend to describe the meanings of the green line and the blue line.
  9. The second white-color peak in the top plot of Figure 16 is not symmetrical. What is the reason? Whether it will affect the thickness measurement accuracy?
  10. The authors introduced the measurement method in detail. But I think the evaluation for measurement performance is not sufficient. I recommend the authors supplement some experiments to quantificationally evaluate the repeatability of their instrument at various angles.

Author Response

We thank the associate editor and the reviewer for the constructive comments. Here you can find in bold the reply to each question.
The corrections and additions to the text are in RED color.

REVIEWER 1
The authors proposed an instrument to measure the thickness of blown films by using white light interferometry. The film thickness is demodulated with the position difference of the two interferometric peaks that correspond to the reflected light from the film's front and rear surface. The peak position is determined by the position of its envelope maximum. This instrument is very useful in the plastic film manufacturing factory. Before published in MDPI Photonics, I recommend the authors respond to the comments and questions below.
1.    Figure 1 is a photo of a bubble for plastic film extrusion. What's the size of it? I recommend the authors provide the scale of the photo.
We added the bubble size in the caption and in the text.

2.    The authors should explain the acronyms again in the figure caption of Figure 3, i.e. SLED, PST SM 1/2, DAQ, PC.
We added the acronyms explanation in the caption.

3.    Is the modulating wave generated by an arbitrary wave generator? What's the function expression of the wave? In addition, what is the meaning of the y-axis label of Figure 6?
The modulating wave is generated by the analog output of the microcontroller (written in line 139), it is a triangular wave numerically smoothed and we don’t have an analog function. LSB is the number of Less Significant Bit (the number read by the microcontroller to generate the wave). A comment was added in the text.

4.    Figure 7 and Figure 8 show the screenshots of the graphic user interfaces of a Tek oscilloscope (OSC) and frequency spectrum analyzer (FSA), respectively. I think it is OK for a thesis paper. But we seldom put software screenshots in a journal article. I don't have the same OSC and FSA as yours. So, I cannot easily read out the amplitudes of the wave or the frequencies of the harmonics. Actually, Figures 12, 15, and 16 have a similar problem.
We added the scales in the captions for all the figures.

5.    Figure 9 is too blurry and omits the plastic film under test. Thus, the authors avoided answering the problem that how to assure the incident angle of the light is perpendicular to the film surface.
We replaced figure 9 with a higher quality one, with the film indicated. We added a comment on the incident angle: “Another factor to be considered is the sensitivity to the incident angle of the light, that in theory should be perpendicular to the film surface. With a focal length of 25 mm and a collimated beam diameter of 1.4 mm there are ±1.6 degrees of tolerance for the alignment, to still get some reflected signal. The error in thickness measurement for this kind of misalignment is absolutely negligible (it is a cosine error): the problem related to misalignment is only signal fading; if the sensor can see the signal the measurement value is correct.

6.    The peak amplitudes in Figure 13 are not consistent with those in Figure 12. From the light blue signal in Figure 12, we can see that the amplitudes of the first and the fourth peaks are approximately equal and the amplitudes of the second and the third peaks are approximately equal. However, we cannot see this phenomenon in Figure 13.
The peaks amplitude depends on the angular alignment of the sensor, on the parallelism of the two surfaces and also on the absolute distance (the relative position with respect to the focus). For these reasons, in every measurement we can see different amplitudes, also relative between the two surfaces. A comment was added in the text.

7.    Figure 14 should have a legend to describe that the blue dot is measured data while the red line is the fitting result.
We added the description in the figure caption.

8.    Figure 16 should have a legend to describe the meanings of the green line and the blue line.
We added the description in the caption: the two vertical lines indicates the position of the peaks, localized by the parabolic regression algorithm.

9.    The second white-color peak in the top plot of Figure 16 is not symmetrical. What is the reason? Whether it will affect the thickness measurement accuracy?
Figure 16 shows a measurement acquired on a real bubble, vibrating. The asymmetry is quite normal for these measurements, because the vibration still changes a little the peak position (the scanning speed is much higher than the vibration speed, but there is still an influence). By averaging 10 measurement we get normally a peak that is not perfectly symmetric. The evaluation of the measurement uncertainty due to this effect was made on real plant. As reported in the discussion, the standard deviation in laboratory condition is about 0.1 µm, while on real bubble becomes about 1-2 µm. A comment was added in the text.

10.    The authors introduced the measurement method in detail. But I think the evaluation for measurement performance is not sufficient. I recommend the authors supplement some experiments to quantificationally evaluate the repeatability of their instrument at various angles.
As expected from theory, we cannot see any measurement error due to the incident angle, because the angle tolerance is about ±1.6 degrees for the alignment, to still get some reflected signal: the error is a cosine function of the angle and it is negligible for this misalignment (lower than 10-3). The main problem of misalignment is signal fading: on real bubble the sensor is carefully aligned to be perpendicular to the film surface, by looking at the signal amplitude. We experimentally see that the bubble vibrations induce rapid distance variations, but small angular misalignment. On tens of measurement we have noticed no signal fading due to misalignment.
A comment was added in the discussion, to better explain the misalignment contribution and the results of different measurement campaign, made in laboratory and on field.

Reviewer 2 Report

The authors reported a low-coherent interferometry-based sensor system for non-contact thickness measurement of blow films. The reported technique could be interesting to the professional group. The authors might want to consider the following comments.

  1. The authors might want to consider including more information for Fig. 6, which is quite confusing, especially the label for the y-axis.
  2. On page 5, the authors claim that “Considering then that the refractive index of the optical fiber is equal to 1.45, there is an effective variation of the optical path of 5.51 μm for each applied volt”. The authors should be cautious making this statement, because when the fiber is stretched, the effective refractive index decreases correspondingly. The round-trip path should be considered here as well.
  3. In Figs. 3 and 11, the abbreviations should be defined.
  4. The authors might have to label the input ports and outputs ports of the commercial Michelson interferometer and include the labels in Figs. 3 and 11 to make the system clear.

Author Response

We thank the associate editor and the reviewer for the constructive comments. Here you can find in bold the reply to each question.

The corrections and additions to the text are in RED color.

REVIEWER 2

The authors reported a low-coherent interferometry-based sensor system for non-contact thickness measurement of blow films. The reported technique could be interesting to the professional group. The authors might want to consider the following comments. 

  1. The authors might want to consider including more information for Fig. 6, which is quite confusing, especially the label for the y-axis.

We added the explanation in the figure caption.

  1. On page 5, the authors claim that “Considering then that the refractive index of the optical fiber is equal to 1.45, there is an effective variation of the optical path of 5.51 μm for each applied volt”. The authors should be cautious making this statement, because when the fiber is stretched, the effective refractive index decreases correspondingly. The round-trip path should be considered here as well.

We agree with the reviewer, that’s why the system needs a calibration, shown in figure 14. We added a comment in the text to underline this concept.

  1. In Figs. 3 and 11, the abbreviations should be defined.

We added the acronyms definition in the captions.

  1. The authors might have to label the input ports and outputs ports of the commercial Michelson interferometer and include the labels in Figs. 3 and 11 to make the system clear.

We added labels to each port in figures 3, 4 and 11, described in the text and figure captions.

Reviewer 3 Report

  1. Because the measurement distance is less than 100 um, please add the calibration results of the target distance around hundreds um. 
  2.  Please add the X and Y-axis units of the plots in Figure 15a&b. 

Author Response

We thank the associate editor and the reviewer for the constructive comments. Here you can find in bold the reply to each question.

The corrections and additions to the text are in RED color.

REVIEWER 3

  1. Because the measurement distance is less than 100 um, please add the calibration results of the target distance around hundreds um. 

We understand that the measurement distance was not properly described, because the film thickness is typically in the order of 100 µm, while the working distance is the focus distance of the second lens of figure 9, equal to 25 mm. The sensor can work from about 23 mm to about 27 mm, and the calibration of the sensor is reported for this range in figure 14.

A better description was added in the text, and in the captions of figure 9 and 14.

  1.  Please add the X and Y-axis units of the plots in Figure 15a&b. 

We added the scales in the captions for the two figures.

Round 2

Reviewer 1 Report

The authors responsed to all my comments and questions. I recommend the revised manuscript to be published on the Photonics.